# Increased Zygote-Derived Plantlet Formation through *In Vitro* Rescue of Immature Embryos of Highly Apomictic *Opuntia ficus-indica* (Cactaceae)

**DOI:** 10.3390/plants12152758

**Published:** 2023-07-25

**Authors:** Angela Carra, Caterina Catalano, Ranjith Pathirana, Maurizio Sajeva, Paolo Inglese, Antonio Motisi, Francesco Carimi

**Affiliations:** 1CNR—Istituto di Bioscienze e BioRisorse, Via Ugo La Malfa 153, 90146 Palermo, Italy; angela.carra@ibbr.cnr.it (A.C.); caterina.catalano@ibbr.cnr.it (C.C.); antonio.motisi@ibbr.cnr.it (A.M.); 2Plant & Food Research Australia Pty Ltd., #46 Plant Breeding, Waite Road, Urrbrae, SA 5064, Australia; ranjith.pathirana@adelaide.edu.au; 3School of Agriculture, Food and Wine, University of Adelaide, Waite Road, Urrbrae, SA 5064, Australia; 4Department of Biological, Chemical and Pharmaceutical Sciences and Technologies (STEBICEF), University of Palermo, Via Archirafi 18, 90123 Palermo, Italy; maurizio.sajeva@unipa.it; 5Department of Agricultural Food and Forest Sciences, University of Palermo, Viale delle Scienze, 90128 Palermo, Italy; paolo.inglese@unipa.it

**Keywords:** apomixis, embryo rescue, ovule culture, prickly pear, cactus pear, nucellar embryos, hybrid progeny

## Abstract

*O. ficus-indica* (prickly pear cactus) is an important forage and food source in arid and semiarid ecosystems and is the most important cactus species in cultivation globally. The high degree of apomixis in the species is a hindrance in plant breeding programs where genetic segregation is sought for the selection of superior genotypes. To understand if *in ovulo* embryo rescue could increase the proportion of zygotic seedlings, we compared the mature seed-derived seedlings with those regenerated from *in vitro* embryo rescue at 20, 25, 30, 35, and 40 post-anthesis days (PADs) in four Italian cultivars. The seedlings were classified as apomictic or zygotic based on molecular marker analysis using inter-sequence single repeat (ISSR) primers. Multiple embryos were recovered from all the cultured immature ovules, and plantlets were regenerated and acclimatized to the field post hardening, with success rates ranging from 62% (‘Senza spine’) to 83% (‘Gialla’). The level of polyembryony differed among cultivars and recovery dates, with the highest being ‘Rossa’, producing 4.8 embryos/ovule at 35 PADs, and ‘Gialla’, the lowest, with 2.7 at 40 PADs. The maximum number of embryos observed within a single ovule was 14 in ‘Trunzara bianca’. ISSR analysis revealed that ovule culture at 35 PADs produced the highest percentage of zygotic seedlings in all the cultivars, from 51% (‘Rossa’) to 98% (‘Gialla’), with a high genotype effect as well. Mature seeds produced much fewer seedlings per seed, ranging from 1.2 in ‘Trunzara bianca’ to 2.0 in ‘Rossa’ and a lower percentage of zygotic seedlings (from 14% in ‘Rossa’ to 63% in ‘Gialla’). Our research opens a pathway to increase the availability of zygotic seedlings in *O. ficus-indica* breeding programs through *in ovulo* embryo culture.

## 1. Introduction

Prickly pear (*Opuntia ficus-indica* (L.) Mill.), also called cactus pear, Barbary fig, or nopal cactus, is a member of the *Cactaceae* family; it originated in central Mexico and is considered the most important cactus species in horticulture worldwide, with a global distribution [1,2]. Photosynthetic adaptation with Crassulacean acid metabolism (CAM), where carbon is fixed in the night when the air is cooler, allows *Opuntia* spp. to better adapt to conserve water in arid or semiarid environments than the C4 and C3 plants, with 3–5 times lower transpiration rates [2,3]. Its excellent adaptation to arid and semiarid climates makes its fruit and fresh stems (cladodes) an important source of human nutrition as well as a forage and fodder source for farm animals in such areas [1,2,4].

The fruits of *O. ficus-indica* cultivars, known as prickly pears or tunas, can be very sweet. They come in diverse colors and are highly appreciated in many cultures [1,5,6]. For a long time, Mexicans have used the tender young cladodes (nopalitos) as a source of green and fresh vegetables [2,7]. Although nopalitos are not a common food in industrialized countries [2,8], they are gaining popularity among European and US consumers for their health benefiting profile. Furthermore, the prickly pear has many medicinal properties and has been used in traditional Mexican medicine for the treatment of a variety of diseases, such as arteriosclerosis, diabetes, and gastritis [2,9,10]. Additionally, the large amount of biomass produced by the prickly pear due to its high shoot-to-root ratio combined with its high productivity makes it an ideal fodder and feed source for livestock in semiarid and arid climates. A five-year study comparing the response of three *Opuntia* species to irrigation in Logandale, Nevada, USA showed that *O. ficus-indica* is the preferred species in terms of biomass gain and fruit quality [11]. Global prickly pear production is dominated by three countries: Mexico (45%), Italy (12.2%), and South Africa (3.7%). In Mexico, the planted area is between 50,000 and 70,000 ha, with an annual production of 300,000–500,000 tons [5,9,12]. Cultivation in Italy is concentrated in Sicily with about 8600 hectares, ranking first among the Mediterranean regions for producing and exporting prickly pear fruits [13].

Thanks to a mix of reproductive strategies (sexual, apomictic, and other clonal strategies), reduced water loss due to succulent stems (including many other morphological and physiological features), and efficient photosynthesis through CAM and polyploid genomes, opuntioid genera have successfully established in many parts of arid and semiarid ecologies, although they are endemic to the Americas [14,15,16]. For example, on the planet’s driest continent, Australia, all opuntoid cacti present (*Austrocylindropuntia* spp., *Cylindropuntia* spp., and *Opuntia* spp.), except *O. ficus-indica*, were named as Weeds of National Significance in April 2012 [17]. *Opuntia* and *Cylindropuntia* are the most invasive among cacti in Australia, with tiger pear (*O. aurantiaca*), prickly pear cactus (*O. monacantha*), wheel cactus (*O. robusta*), white-spined prickly pear (*O. streptacantha*), common prickly pear (*O. stricta*), and velvet prickly pear (*O. tomentosa*) being the most widespread [18]. South Africa and Spain are also considered invasive hotspots for opuntoid cacti [14]. These facts illustrate the high degree of adaptation of *Opuntia* spp. to drier regions of the world where other crops struggle, and the need to exploit it in horticulture in these regions.

Exploiting apomixis is a natural way of cloning through seeds, as apomixis produces seedlings that are genetically identical to the mother plant without the involvement of male gametes [19,20]. Apomixis is a complex developmental process; it is historically subdivided into two categories, gametophytic and sporophytic, based on whether the embryo develops via a gametophyte (embryo sac) or directly from a diploid somatic (sporophytic) cell within the ovule [20]. Apomixis in angiosperms is rarely obligate; usually, apomictic plants produce asexual and sexual progeny within the same offspring generation, and asexuality is facultative. Therefore, a proportion of the offspring represents recombinants, but frequencies of sexuality vary a lot among genera, species, and different modes of apomixis [21,22]. Diplospory is a form of gametophytic apomixis in which an unreduced embryo sac forms from a megaspore mother cell with the circumvention of meiosis [20,23]. Apomixis has a genetic basis, but it is still a matter of question how it is regulated. The ability to produce genetically uniform progeny via seeds is of significant value for its potential in agriculture to fix complex favourable genotypes, particularly hybrids expressing heterosis or those obtained from wide crosses, to improve breeding programs’ efficiency [19,20,21,23].

On the other hand, the apomictic embryos, which are genetically identical to their maternal parent, limit the range of genetic variability that can be observed in the progeny of a cross, and thus the possibility of finding new genotypes. In these cases, *in vitro* embryo rescue can be a very useful technique for breeding programs [24,25].

Apomixis frequently occurs in *Opuntia* spp., including *O. ficus-indica* [15,19]. It was initially described by Ganong in *O. vulgaris* as far back as 1898 [20]. The most common type of apomixis in *Opuntia* involves the development of adventitious embryos from nucellar tissue (sporophytic agamospermy) [21,22,23]. Or, as in *O. streptacantha*, embryos can develop from an unfertilized egg (diplospory parthenogenesis) [21]. More recently, Kaaniche-Elloumi et al. [24] reported that *O. ficus-indica* ovules showed both sporophytic and gametophytic embryogenesis.

The objectives of this research were to study apomixis in *O. ficus-indica* and to determine the incidence of sexual and apomictic embryos *in vitro* and *in vivo*. In order to determine whether the genotype influences the level of polyembryony, four different cultivars were used. We also analyzed the effect of the ovule isolation time on the proportion of sexual and apomictic offspring *in vitro*, with the aim of developing a protocol to increase the production of zygotic seedlings in crossbreeding programs.

## 2. Materials and Methods

### 2.1. Plant Material

To study polyembryony, four cultivars used for fresh fruit consumption were utilized: ‘Rossa’, ‘Senza spine’, ‘Trunzara bianca’, and ‘Gialla’. All the genotypes analyzed are classified as facultative apomicts because they have the ability to reproduce both sexually and asexually through apomixis. Immature and mature fruits were collected from open pollinated adult plants growing at the germplasm repository for perennial plants at the Institute of Biosciences and BioResources of the National Research Council of Italy (CNR-IBBR), located in Collesano District (Province of Palermo), Italy (37°59′19.9″ N, 13°54′55.8″ E, 80 m a.s.l.). Fruit growth and development are shown in Figure 1 and Appendix A.

### 2.2. Media and In Vitro Culture Methodology of Immature Ovules

Media preparation, culture conditions, and plant regeneration were conducted similarly to the methods described by Carimi et al. [25]. Fertilized ovules to be grown *in vitro* were excised from immature fruits collected at 5-day intervals from 20 to 40 post-anthesis days (PADs) during July and August. The immature fruits were rinsed with tap water and then surface-sterilized by immersion for 5 min in 70% ethanol and 30 min in 2% (*w*/*v*) sodium hypochlorite. Finally, the fruits were rinsed two times with sterile distilled water for 5 min under aseptic conditions (Figure 2A,E). After sterilization, one longitudinal cut was made immediately under the fruit epidermis, avoiding the core where fertilized ovules are embedded. After opening the immature fruits (Figure 2B,F), fertilized ovules were extracted under aseptic conditions (Figure 2C,G), and, by means of a longitudinal cut, the outer seed integument was removed under a stereoscopic microscope using a scalpel and forceps (Figure 2D,H).

Immature ovules without integument were cultured on plant growth regulator-free Murashige and Skoog [26] (MS) basal medium (micro and macro salts and MS vitamins) supplemented with sucrose (50 g L^−1^) and 500 mg L^−1^ malt extract and solidified with 7 g L^−1^ Plantagar (S 1.000, B&V, Italy). To induce embryo development and to determine the percentage of responsive ovules, each immature ovule was placed on 8 mL of medium in plastic Petri dishes (60 × 15 mm) sealed with Parafilm M (Figure 3A). Cultures were maintained in a climatic chamber at 26 °C with a 16 h photoperiod (40 μmol m^−2^ s^−1^ at shelf level provided by Osram Cool White 18 W fluorescent lamps).

Four weeks after incubation, each embryo sac was scored for the presence or absence of one or more embryos. About six weeks after incubation, the embryos generated from immature ovules were collected and transferred to solid MS medium prepared as previously described in Petri dishes (100 × 20 mm) and cultured for a further 4–6 weeks to allow plantlet development. Individual germinated somatic embryos (about 1–2 cm in length) were transferred to Magenta™ vessels to allow further growth (one embryo/Magenta™ vessel containing 50 mL of basal MS medium).

Once rooted, plantlets were transferred to autoclaved Jiffy^®^ peat pellets and maintained for five weeks in a basal heating bench at 25 °C and at high relative humidity (95–98%). Subsequently, the plants were pricked into pots containing sterile soil, transferred to the greenhouse, and exposed to natural daylight conditions at 22/27 °C night/day.

### 2.3. Seed Germination In Vivo

Fresh seeds were collected from mature fruits harvested in September stratified at 4 °C for 3 months in the dark and germinated into plastic pots (70 mm × 70 mm) containing sterile soil. The potted plants, covered with transparent polyethylene bags to maintain temperature and high humidity, were placed in a climate chamber at 25 ± 1 °C under the same culture conditions as described above. The percentage of germination and the number of plantlets produced per seed were evaluated four months after sowing.

### 2.4. DNA Extraction

DNA was extracted from young cladodes of the mother plants growing in the field and from young seedlings regenerated from ovules *in vitro* and from seeds *in vivo.* Seedlings from different cultivars were randomly selected from each different ovule isolation time for the analyses of genetic origin (zygotic or apomictic). All the samples were frozen in liquid nitrogen and stored at −80 °C. They were ground in a mortar with liquid nitrogen, and genomic DNA was extracted using the procedure described by Doyle and Doyle [27]. DNA was quantified by measuring OD_260_, as described by Sambrook et al. [28].

### 2.5. Genetic Analysis

To assess the genetic origin of the progeny, mother plants and plantlets generated from ovules *in vitro* and from seeds *in vivo* were characterized by inter-simple sequence repeat polymorphic DNA (ISSR) marker analysis, as described by Siragusa et al. [29]. Briefly, a total of ten primers as reported by Fang and Roose [30], were used in preliminary experiments to assess the genetic origin of seedlings. Five of those primers, i.e., (AC)_8_YG, (AC)_8_YA, (TCC)_5_RY, (GA)_8_YC, and (GA)_8_YG were low informative and therefore were not included in the final study. The primers used in the final study were (AG)_8_YC [Annealing Temperature (Ta) 52.6 °C], (AC)_8_YT (Ta 50.3 °C), (AG)_8_YT (Ta 50.3 °C), (GT)_8_YG (Ta 52.1 °C), and (CA)_8_RG (Ta 51 °C). To distinguish apomictic from zygotic seedlings, a genetic analysis based on ISSR analysis was performed, as previously described in detail [29]. To confirm the reproducibility of the banding patterns, all analyses were repeated twice.

### 2.6. Statistical Analysis

The fruit growth pattern was evaluated by measuring fruit fresh weight, diameter, and length of 20 fruits for each cultivar at 20–40 PADs at 5-day intervals and at the ripening stage.

Each treatment *in vitro* and *in vivo* comprised 60 ovules or seeds, and experiments were performed in triplicate in a randomized complete block design. The effects of genotype and ovule developmental stage on the percentage of responsive ovules, the average number of plantlets generated per ovule, the percentage of monoembryonic ovules, and the percentage of ovules and seeds producing zygotic seedlings were tested by ANOVA (*p* ≤ 0.05), and the differences among means were tested by Tukey’s test. Prior to analysis, percentage data were arcsin square root transformed. Statistical analysis was performed using SigmaStat 3.5 for Windows.

## 3. Results

### 3.1. Embryo Rescue In Vitro

The first embryos emerged from the immature ovules cultured on MS medium (Figure 3A) about one week after culture initiation, and after 3–4 weeks, several embryos were visible on the surface of the ovule (Figure 3B). Fertilized ovules contained several embryos at different developmental stages (Figure 3C). The maximum number of viable embryos observed in a single fertilized ovule varied according to the cultivar: ‘Senza spine’ had a maximum of eight, ‘Rossa’ had a maximum of twelve, ‘Gialla’ had a maximum of twelve, and ‘Trunzara bianca’ had a maximum of fourteen.

A high percentage of embryos germinated *in vitro* (Figure 3D), and about 8–10 weeks after culture initiation, the plantlets grew normally (Figure 3E) with no significant differences found among the different genotypes. After about 3–4 months of culture *in vitro,* the quality of the roots was good, and plantlets were transferred to Jiffy peat pellets (Figure 3F). The percentage of acclimatized plantlets observed for the different cultivars was: 62%, 71%, 75%, and 83% for ‘Senza spine’, ‘Trunzara bianca’, ‘Rossa’, and ‘Gialla’, respectively.

Responsive ovules were collected from all genotypes at different PADs. The percentage of responsive explants ranged from 10% (‘Gialla’ collected at 40 PADs) to 97% (‘Senza spine’ collected at 35 PADs). The best result for all cultivars was obtained when collection was performed at 35 PADs (Figure 4).

The percentage of responsive ovules varied according to the collection time. The value increased when ovules were isolated in the period lasting from 20 to 35 PADs, while it decreased significantly at 40 PADs (Figure 4).

No significant differences were found among cultivars for the number of plantlets per ovule, while the number of plantlets per ovule was significantly lower for recovery at 40 PADs when compared to earlier periods of embryo rescue. Overall, data attest their value between 3.6 (‘Gialla’) and 4.6 (‘Rossa’) plantlets per ovule when genotype is considered, while data ranged between 3.5 (40 PADs) and 4.8 (35 PADs) for the number of plantlets regenerated per ovule according to time of recovery (Figure 5).

The percentage of monoembryonic ovules varied greatly in the experiment (Figure 6). The highest percentage was achieved with ‘Trunzara bianca’ collected at 40 PADs (22.9%) and the lowest percentage was recorded with the same cultivar collected at 20 PADs (5.0%). However, no significant differences were found among cultivars; percentages ranged from 13.4% (‘Rossa’) to 17.7% (‘Trunzara bianca’). With regard to the time of recovery, no significant differences in percentages of monoembryonic ovules were observed.

Ovules versus seeds: The number of plantlets generated per ovule *in vitro* and per seed *in vivo* varied greatly, and the number generated from ovules was significantly higher than those from seeds for all four cultivars (Figure 7). The average number of plantlets obtained per ovule ranged from 3.64 to 4.62 (‘Gialla’ and ‘Rossa’, respectively), with no significant differences among cultivars. Conversely, the average number of plantlets obtained from seeds *in vivo* was strongly reduced (Figure 7), ranging from 1.21 to 1.99 (‘Trunzara bianca’ and ‘Rossa’, respectively).

### 3.2. Genetic Analysis

ISSR primers were used to amplify the DNA of regenerants from each cultivar and to compare them to the respective mother plant. The presence of polymorphic bands allowed us to detect zygotic and apomictic seedlings (Figure 8).

Screening using molecular markers revealed that a genotype had a significant effect on the percentage of ovules producing zygotic seedlings (Figure 9). The highest percentage was achieved with ‘Gialla’ collected at 35 PADs (98%) and the lowest percentage was recorded with ‘Rossa’ collected at 20 PADs (33%). Significant differences were also observed among cultivars; percentages ranged from 42.2% (‘Rossa’) to 92.4% (‘Gialla’). However, no significant differences were found among different ovule isolation times; percentages ranged from 58.5% (20 PADs) to 74.5% (35 PADs).

Ovules versus seeds: From our results, it appears that the *in vitro* ovule culture procedure allows a more efficient recovery of zygotic embryos than the traditional *in vivo* seed germination procedure (Figure 10). The percentage of ovules with a zygotic seedling was higher than and significantly different from the values for seeds with a zygotic seedling in all the cultivars, ‘Trunzara bianca’ and ‘Gialla’ being the most responsive. The highest percentage was achieved with ovules of ‘Gialla’ (92.40%), and the lowest percentage was recorded with seeds of ‘Rossa’ (13.78%).

## 4. Discussion

Under the rainfed conditions of the semiarid highlands of central Mexico, cactus pear (*O. ficus-indica*) is the main fruit crop, with more than 50,000–70,000 ha planted [5,22]. Cactus pear is also important in Italy and South Africa [12]; it is gaining importance in Chile [31], Brazil [32], and Egypt [10]; and it is becoming an important alternative crop for several countries in North Africa and other semiarid areas of the world [2,3,4,33,34]. Currently, in all countries with commercial plantations, the crop is produced from a few varieties that have either a direct origin in Mexico or have been derived from those [15,35]. The narrow nature of the germplasm base in Italy is also evident from our results of fruit characteristics, as there were no statistical differences in fruit length, diameter, or weight among the four studied cultivars. Producing varieties with better adaptation to the local environment, resistance to disease, and improved fruit or forage quality is an important objective in cactus pear breeding programs [15,35]. Climate change, while adding more opportunities for cactus cultivation in new areas, will require the achievement of other novel objectives in breeding programs [36,37]. Among the reproductive strategies evolved in *Opuntia* spp., apomixis and vegetative propagation by cladode detachment can be used for clonal propagation, and these are valuable tools for breeders and nurseries. This is the main reason for the lack of genetic diversity in *O. ficus-indica.* While the prevalence of apomixis in *O. ficus-indica* gives an additional tool for the nursery industry for vegetative propagation of elite genotypes, the identification of hybrids and progeny selection in crossbreeding programs becomes challenging, complicated, and inefficient because of apomixis [22].

Embryo rescue is a biotechnological approach used to overcome some technological difficulties encountered when using traditional plant breeding approaches. Early rescue of hybrid embryos allows the recovery of interspecific and intergeneric hybrids that are impossible to produce *in vivo* [38,39]. The method is also used to manipulate ploidy in cultivated species [38,40]. In this study, we explored another possible application of embryo rescue, i.e., to enhance the regeneration of zygotic embryos in *O. ficus-indica.* Previously, embryo rescue has been employed to increase the ratio of zygotic embryos to apomictic embryos in other apomictic species, such as citrus [25,41]. In cacti, Felker et al. [42] tested the progeny of a cross between *O. lindheimerii* and *O. ficus-indica* using randomly amplified polymorphic DNA (RAPD) markers and confirmed that four out of thirteen (30.8%) tested progeny were apomicts. In our study, we show conclusively that within *O. ficus-indica,* this ratio is genotype-dependent, with just 13.8% of the seedlings being zygotic in ‘Rossa’ against 63.3% in ‘Gialla’ in seedlings grown from mature seeds. While there are several mechanisms involved in apomixis, many studies have confirmed it to be controlled as a dominant trait [43,44], and the complexities unravelled in molecular studies can be attributed mainly to secondary factors resulting from the reproductive process [45,46]. Therefore, induced mutagenesis could be used to produce non-apomictic genotypes to help in crossbreeding programs [47,48]. However, a more urgent need is the development of methods to recover more zygotic seedlings from existing cultivars in crossbreeding programs. Our research was directed at a solution to solve this problem.

We attempted embryo rescue over five PADs periods from 20 to 40 days in four Italian cultivars with contrasting morphologies, and in all the cultivars, 35 PADs embryo rescue was the most successful in terms of the percentage of responsive ovules (80–95%) and the mean number of responsive embryos per ovule (4.4–5.3) in all four cultivars. In contrast, mature seeds produced very low numbers of seedlings per seed (1.2–2, or almost fourfold less). Our method of acclimation and hardening of immature embryo-derived seedlings was efficient, and the success rate was from 62% (‘Senza spine’) to 83% (‘Gialla’). The next step in our research was to identify the origins of the seedlings, and we used ISSR markers, which are highly efficient and reliable [49,50,51]. Again, we found genotypic effects on the percentage of seedlings of zygotic origin, with ‘Rossa’ and ‘Senza spine’ producing significantly less (42 and 46%, respectively) than ‘Trunzara bianca’ and ‘Gialla’ (85 and 92%, respectively). Importantly, our method yielded a significantly higher percentage of zygotic embryo-derived seedlings than the counterpart mature seeds (14–63%) in all four cultivars. It should be noted that this higher percentage of zygotic embryos in our *in vitro* approach is from a fourfold higher ovule response compared to mature seeds, as already noted, thus making the yield of zygotic seedlings even greater. Of the four periods tested, 35 PADs recorded the highest yield of zygotic seedlings in all four cultivars, thus making our protocol easy to follow. It appears that for any species, the optimum period for the rescue of embryos needs to be identified, as previously recorded in apomictic sour orange (*Citrus aurantium*—125 PADs) [25] and in ‘Shiranuhi’ mandarin, a hybrid citrus [(*C. unshiu* × *C. sinensis*) × *C. reticulata*] (145 DAP) [41].

In preliminary experiments (data not presented), different combinations of plant growth regulators (PGR) were added to the culture medium to stimulate the *in vitro* development of the zygotic and apomictic embryos present in the immature ovules. Interestingly, we observed that in some combinations, PGR stimulated callus formation and the production of adventitious embryos from the different tissues of immature ovules. On the contrary, the PGR-free medium allowed the regular development of the zygotic and apomictic embryos already present in the immature ovules without the production of callus and adventitious embryos. Therefore, the immature ovules used in the present study were incubated on PGR-free medium to facilitate the recovery of zygotic embryos. The *in vitro* protocol used in our research is simple, as it consists of only MS media supplemented with malt extract and sucrose. For *in ovulo* embryo rescue of *Hylocereus* interspecific hybrids, another cactus of horticultural significance, Cisneros and Tel-Zur [52] used a combination of naphtheleneacetic acid, thidiazuron, and glutamine. In blueberry [38] and gentian [53] *in vitro* ovule culture, casein hydrolysate seems to be an essential ingredient. Thus, a reduced form of organic nitrogen seems to be essential for embryo growth *in vitro*, as also suggested by Sahijram et al. [54].

It is known that in *Opuntia* spp., apomixis can occur mainly through sporophytic agamospermy [21,22,23], where adventitious embryos develop from nucellar tissue. However, the development of embryos from unfertilised ovules (parthenogenesis) has also been observed in the genus *Opuntia* [21]. However, the exact reason for the abortion of zygotic embryos *in vivo* during the seed maturation processes is not known, and we are focusing on this aspect in our current research. Our hypothesis is that the numerous embryos of apomictic origin that are contained in the ovule (often, there are more than ten embryos per ovule) compete with the zygotic embryo by using the resources necessary for its development, causing its abortion. This hypothesis is substantiated by the increase in the proportion and number of zygotic embryos when the immature ovules are incubated on a culture medium providing sufficient nutrients, which enhances the chances of survival of zygotic embryos.

In conclusion, it can be stated that *in ovulo* embryo culture can increase the number of zygotic seedlings and their ratio to apomictic seedlings; therefore, this can play a significant role in crop improvement programs of apomictic *O. ficus-indica* involving hybridisation and selection in segregating populations.

## Figures and Tables

**Figure 1 plants-12-02758-f001:**
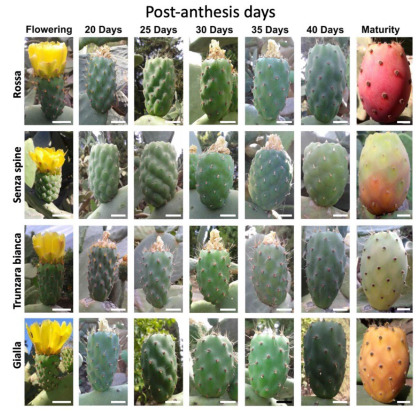
Prickly pear fruits of four cultivars harvested at 5-day intervals from 20 to 40 post-anthesis days for *in vitro* embryo rescue and, at maturity, for *in vivo* germination. Bar = 2 cm.

**Figure 2 plants-12-02758-f002:**
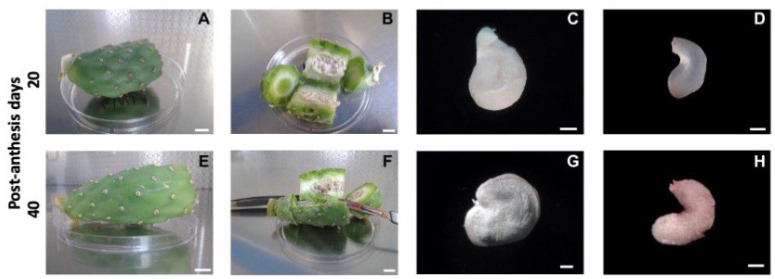
Procedure for ovule dissection under aseptic conditions. (**A**,**E**) Fruits harvested at different post-anthesis days (20 and 40, respectively) were surface sterilized in a laminar flow hood. Bar = 1 cm. (**B**,**F**) Fruits dissected in halves. Bar = 1 cm. (**C**,**G**) Immature ovules dissected from fruit. Bar *=* 1 mm. (**D**,**H**) The ovules after the outer integument was removed with a razor blade. Bar = 1 mm.

**Figure 3 plants-12-02758-f003:**
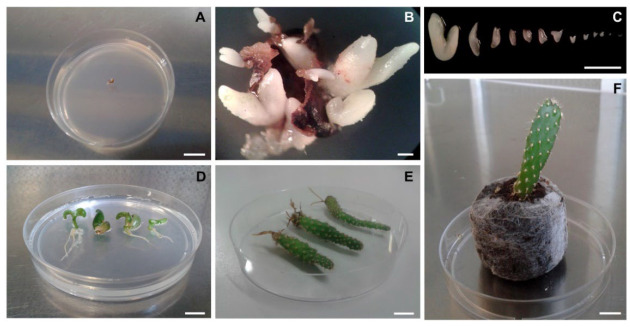
*In vitro* recovery procedure of *Opuntia ficus-indica* embryos and plantlets. (**A**) The fertilized immature ovule, without the outer integument, incubated on MS medium. Bar = 1 cm. (**B**) Embryos arising from fertilized immature ovule incubated *in vitro* on MS medium. Picture taken 3 weeks from the beginning of the experiment. Bar = 1 mm. (**C**) Embryos at different developmental stages dissected from a single fertilized ovule of ‘Trunzara bianca’ after 4 weeks of incubation. Bar = 5 mm. (**D**) Plantlets growing in Petri dish after 8 weeks from the beginning of the experiment. Bar = 1 cm. (**E**) Plants are removed from Magenta vessels and rinsed thoroughly in water to remove traces of medium; ready for transfer to Jiffy pots. Picture taken 14 weeks from embryo germination. Bar = 1 cm. (**F**) *Opuntia* plant acclimatized in a Jiffy pot after 19 weeks from the beginning of the experiment. Bar = 1 cm.

**Figure 4 plants-12-02758-f004:**
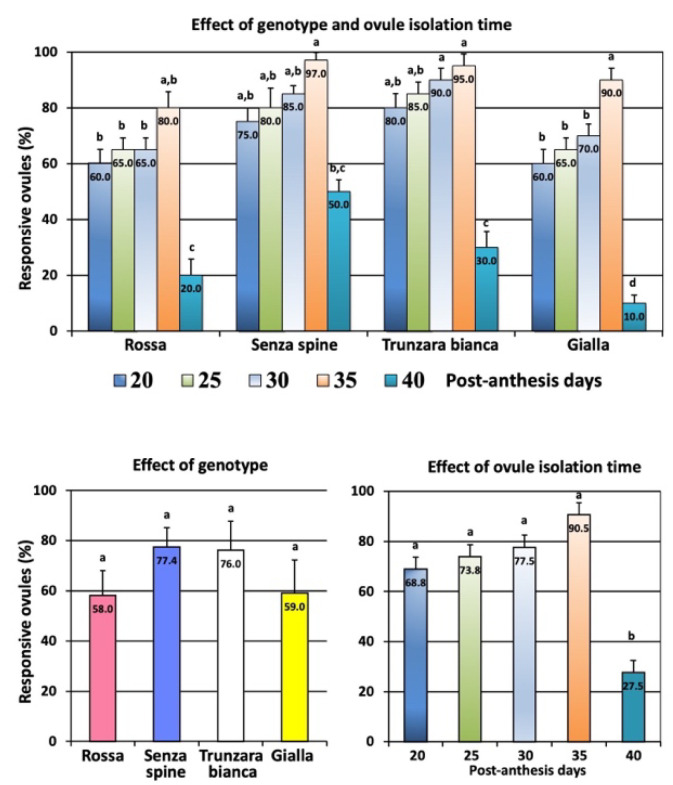
Effect of genotype and ovule isolation time on percentage of responsive ovules incubated *in vitro*. Different letters indicate significant differences (Tukey’s test, *p* < 0.05, n = 60). Data represent values ± SE.

**Figure 5 plants-12-02758-f005:**
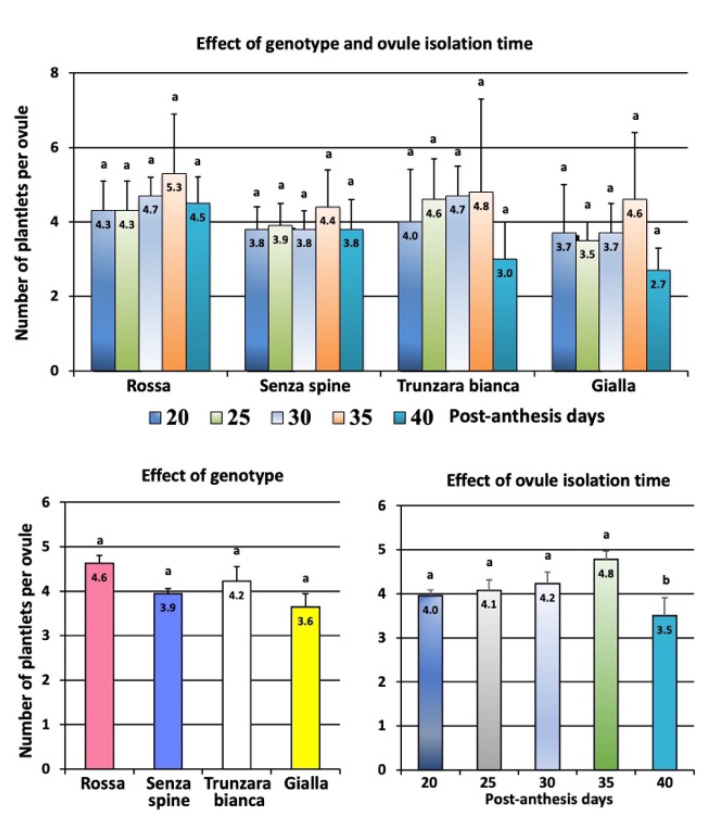
Effect of genotype and ovule isolation time on average number of plantlets generated per ovule. Different letters indicate significant differences (Tukey’s test, *p* < 0.05 level, n = 60). Data represent values ± SE.

**Figure 6 plants-12-02758-f006:**
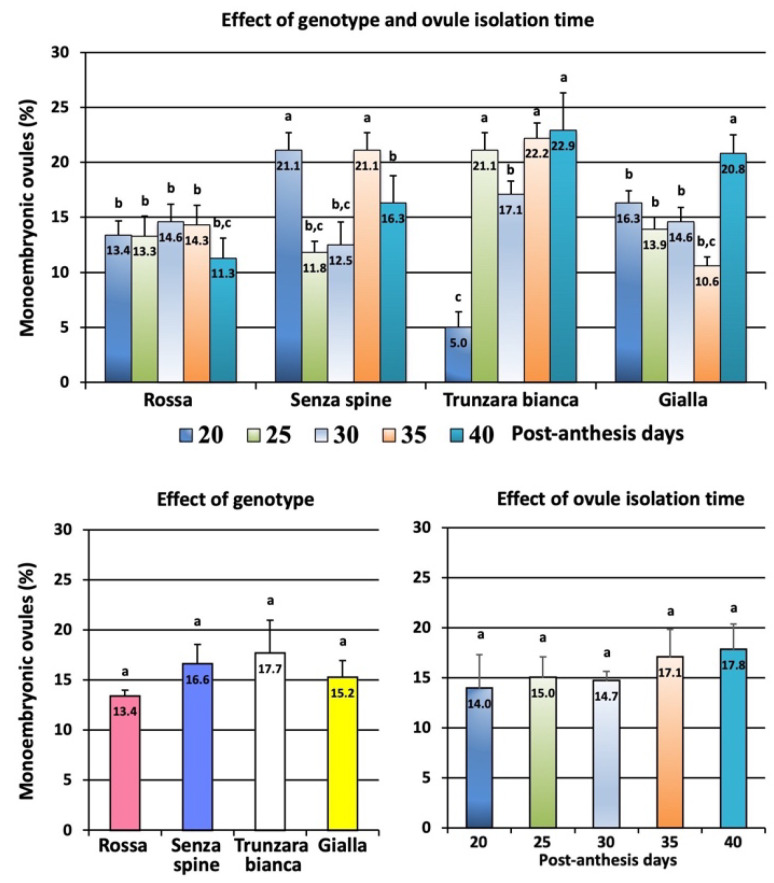
Effect of genotype and ovule isolation time on percentage of monoembryonic ovules. Different letters indicate significant differences (Tukey’s test, *p* < 0.05 level, n = 60). Data represent values ± SE.

**Figure 7 plants-12-02758-f007:**
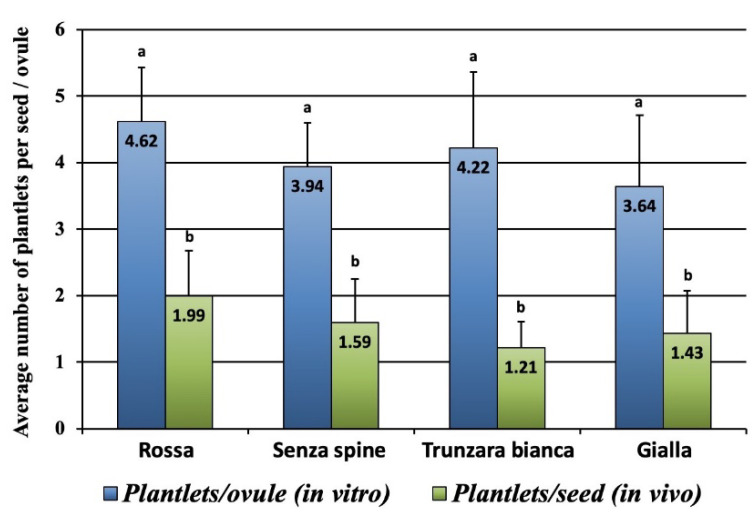
Average number of plantlets generated per ovule *in vitro* and per seed *in vivo*. Different letters indicate significant differences (Tukey’s test, *p* < 0.05 level, n = 60). Bars correspond to mean values ± SE.

**Figure 8 plants-12-02758-f008:**
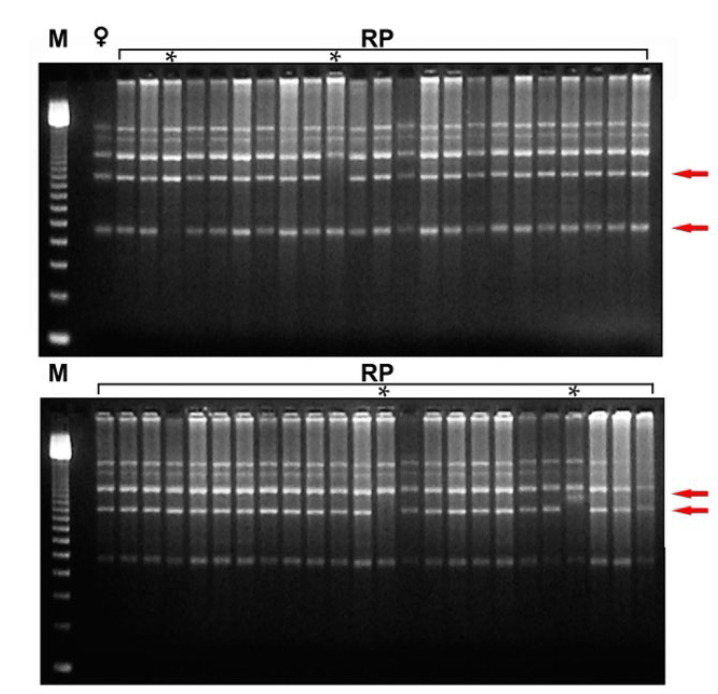
DNA analysis of plantlets recovered *in vitro* from immature ovule culture. Inter-simple sequence repeat polymorphic DNA (ISSR) profiles amplified from DNA extracted from 47 RP of ‘Rossa’ analysed using primer (GT)8 YG. M 100-bp DNA ladder; **♀** mother plant; RP 1–23 and 24–47 plantlets rescued *in vitro*. Arrows indicate polymorphic bands. The asterisk (*) indicates the profiles of the zygotic seedlings.

**Figure 9 plants-12-02758-f009:**
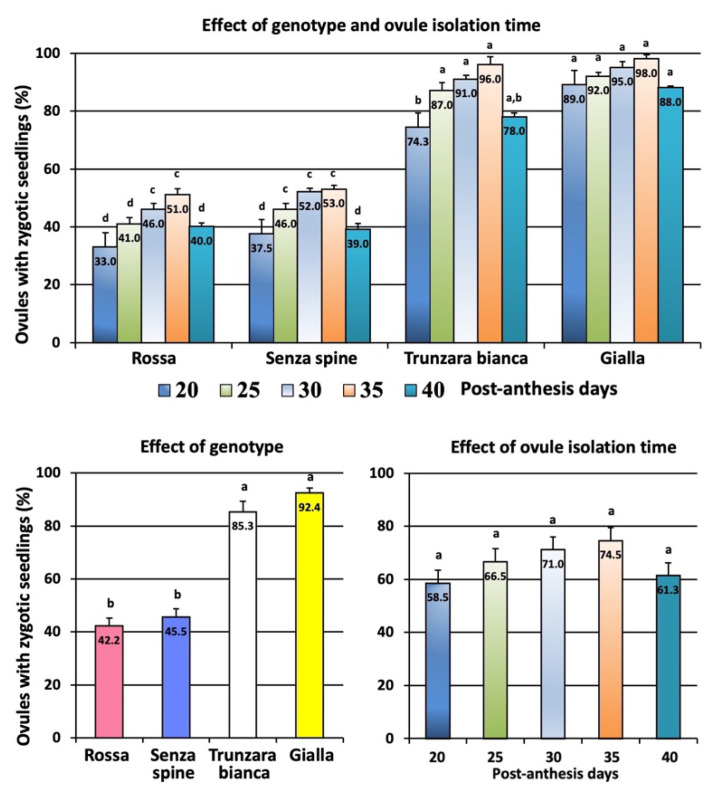
Percentage of ovules, collected at different days post anthesis, that yielded at least one zygotic seedling. Different letters indicate significant differences (Tukey’s test, *p* < 0.05 level, n = 60). Bars correspond to mean percentage values ± SE.

**Figure 10 plants-12-02758-f010:**
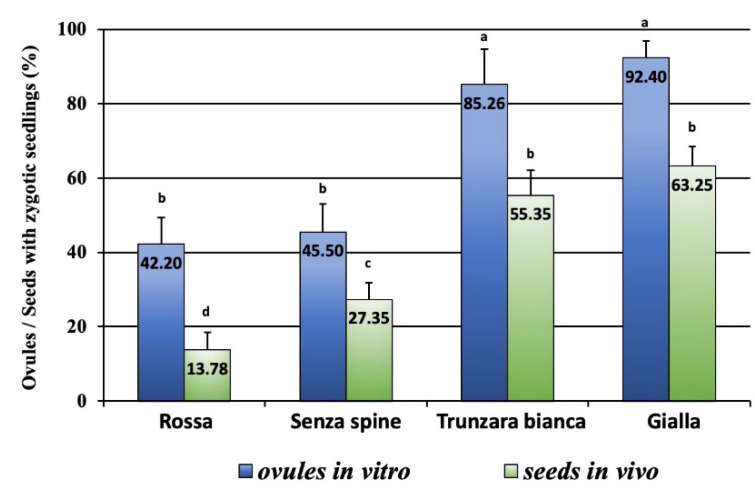
Percentage of ovules and seeds producing zygotic seedlings. Different letters indicate significant differences (Tukey’s test, *p* < 0.05 level, n = 60). Bars correspond to mean percentage values ± SE.

## Data Availability

All data that were generated or analysed during this study have been included in this published article. Statistical data output is available upon request from the corresponding author.

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
