# Peer review of "Increased Zygote-Derived Plantlet Formation through *In Vitro* Rescue of Immature Embryos of Highly Apomictic *Opuntia ficus-indica* (Cactaceae)"

_plants, 2023, doi:10.3390/plants12152758_

Round 1
Reviewer 1 Report
Apomixis is a form of asexual reproduction through seeds. The trait has been reported in more than 300 taxa including the genus Opuntia (Cactaceae). Although the phenomenon is useful in clonal multiplication of elite genotypes, its occurrence is a hinderance in plant breeding programs where genetic segregation of traits is sought for selection
The first paragraph is a scientific definition af Apomixis I think that it will be better if you remove it from the abstract section (can be reintroduced on the introduction is enough)
Figure 1. Bar: 5 cm : Not included for all verities and sometimes the bar is not really clear
Supplementary Figure 1. It will better if you include the acronym of PDA: days post anthesis. To be suitable for Readability it will be better if yoy change the acronym PDA: 1- Post-anthesis days (PADs)
Material and method :
Authors should pay more attention about 2 paragraphs (DNA extraction and Genetic analysis) these 2 section was written quite similar to the reference Siragusa et al., 2007
Siragusa, M., Carra, A., Salvia, L., Puglia, A. M., De Pasquale, F., & Carimi, F. (2007). Genetic instability in calamondin (Citrus madurensis Lour.) plants derived from somatic embryogenesis induced by diphenylurea derivatives. Plant Cell Reports, 26, 1289-1296.
Author Response
Reviewer 1
Q1: Apomixis is a form of asexual reproduction through seeds. The trait has been reported in more than 300 taxa including the genus Opuntia (Cactaceae). Although the phenomenon is useful in clonal multiplication of elite genotypes, its occurrence is a hinderance in plant breeding programs where genetic segregation of traits is sought for selection.
Our comment / answer
We thank the Reviewer for comments. We appreciate the suggestions very much and we will take them into consideration for further improvement of the paper.
Q2: The first paragraph is a scientific definition of Apomixis I think that it will be better if you remove it from the abstract section (can be reintroduced on the introduction is enough).
Our comment / answer
The definition of Apomixis, as suggested by the Reviewer, has been removed from the abstract and is now described in the introduction section. However, to introduce the topic, we had to add another sentence in the abstract.
Q3: Figure 1. Bar: 5 cm: Not included for all verities and sometimes the bar is not really clear.
Our comment / answer
We modified Figure 1 as suggested by the Reviewer by adding the bars in all verities. We also increased the thickness of the bars to make them more visible.
Q4: Supplementary Figure 1. It will better if you include the acronym of PDA: days post anthesis. To be suitable for Readability it will be better if you change the acronym PDA: 1- Post-anthesis days (PADs).
Our comment / answer
We thank the Reviewer for the suggestion. We modified the figures and the text as suggested by the Reviewer by changing the acronym DPA (days after anthesis) with the acronym PDAs (Post-anthesis days).
Q4: Material and method: Authors should pay more attention about 2 paragraphs (DNA extraction and Genetic analysis) these 2 section was written quite similar to the reference Siragusa et al., 2007 (Siragusa, M., Carra, A., Salvia, L., Puglia, A. M., De Pasquale, F., & Carimi, F. (2007). Genetic instability in calamondin (Citrus madurensis Lour.) plants derived from somatic embryogenesis induced by diphenylurea derivatives. Plant Cell Reports, 26, 1289-1296.).
Our comment / answer
The Reviewer's observation is correct, the procedure described in our manuscript for DNA extraction and Genetic analysis is similar to that described in our previous paper (Siragusa et al., 2007). As suggested by the Reviewer we have reduced the description on genetic analysis in the section in M&M and referred to the previous paper instead.

Reviewer 2 Report
I have read the manuscript with pleasure!
Even though I'm no 'English' expert, I think it's written in good, communicative, English. The novelty value is high. Their work should be of great importance both in basic research and in the micropropagation and cultivation of opuntia.
I have few comments. Unfortunately, the authors did not number the verses and it was difficult to mark specific fragments of their manuscript. :/
· Shouldn't the terms in vitro, in vivo be italicised? > in vitro, in vivo ?
· Sometimes authors use the term variety, sometimes cultivar. Were the studied genotypes botanical varieties or cultivars?
· Is prickly pear a self-pollinated or cross-pollinated (self-, cross-fertile) species? Were the mother plants (in germplasm repository) pollinated in a controlled manner or was the pollen donor unknown?
· Fig. 2. Unnecessarily 'amplified' information: 'under aseptic conditions' x3
· Different parts of the text: "between! cultivars" > not: "among cultivars"?
· „No significant differences were observed! between! cultivars” > not: „No significant differences were proven/found among cultivars”?
· Fig.8. Would you like to mark which runs indicate plants from apomixis or from zygotic embryos?
· Please standardize the units: "with sucrose (146 mM)! and 500 mg L–1 malt extract and solidified with 7 g L–1 agar" > with sucrose (50 g L-1)?? and 500 mg L–1 malt extract and solidified with 7 g L–1 agar.
· Please provide more information about the composition of the media. What vitamins were used? What was the difference between the culture medium used during initiation and used during plantlet growth? Did the first one contain PGRs? If so, what are they? Is it possible that the embryos have been cloned during the incubation of the ovules? Was it possible to identify the cloned zygotic embryos?
· Are there any known hypotheses regarding the elimination of zygotic embryos during seed/fruit maturation? Maybe the authors have their own?
Summarizing, in my opinion, the article is valuable and 'almost ready' to be published. :)
Author Response
Reviewer 2
Q1: Even though I'm no 'English' expert, I think it's written in good, communicative, English. The novelty value is high. Their work should be of great importance both in basic research and in the micropropagation and cultivation of opuntia.
Our comment / answer
Thank you for your appreciation and your kind words. We appreciate the suggestions very much and we will take them into consideration for further improvement of the paper.
Q2: Shouldn't the terms in vitro, in vivo be italicised? in vitro, in vivo?
Our comment / answer
All the terms ‘in vivo’ and ‘in vitro’ has been italicised.
Q3: Sometimes authors use the term variety, sometimes cultivar. Were the studied genotypes botanical varieties or cultivars?
Our comment / answer
We thank the reviewer for the remark. The correct term is ‘cultivar’ and the text has been changed accordingly.
Q4: Is prickly pear a self-pollinated or cross-pollinated (self-, cross-fertile) species? Were the mother plants (in germplasm repository) pollinated in a controlled manner or was the pollen donor unknown?
Our comment / answer
We thank the reviewer for comments. The information has been added and the text now sounds as here reported: ‘Immature and mature fruits were collected from open pollinated adult plants growing at the germplasm repository for perennial plants at the Institute of Biosciences and BioResources of the National Research Council of Italy (CNR-IBBR) located in Collesano District (Province of Palermo), Italy (37°59′19.9″ N, 13°54′55.8″ E, 80 m a.s.l.).’
Q5: Fig. 2. Unnecessarily 'amplified' information: 'under aseptic conditions' x3.
Our comment / answer
We thank the reviewer for remark. We modified the legend of Figure 2 as suggested by the Reviewer.
Q6: Different parts of the text: "between! cultivars" > not: "among cultivars"?
“No significant differences were observed! between! cultivars” > not: No significant differences were proven/found among cultivars”?
Our comment / answer
We modified the MS as suggested by the Reviewer.
Q7: Fig.8. Would you like to mark which runs indicate plants from apomixis or from zygotic embryos?
Our comment / answer
We thank the reviewer for the suggestion. We modified the MS following the Reviewer's advice by adding the asterisk (*) in Figure 8 to indicate the profiles of the zygotic seedlings.
Q8: Please standardize the units: "with sucrose (146 mM)! and 500 mg L–1 malt extract and solidified with 7 g L–1 agar" > with sucrose (50 g L-1)?? and 500 mg L–1 malt extract and solidified with 7 g L–1 agar.
Our comment / answer
We modified the MS following the Reviewer's advice.
Q9: Please provide more information about the composition of the media. What vitamins were used? What was the difference between the culture medium used during initiation and used during plantlet growth? Did the first one contain PGRs? If so, what are they?
Our comment / answer
We modified the MS following the Reviewer's advice by adding the requested information.
Q10: Is it possible that the embryos have been cloned during the incubation of the ovules? Was it possible to identify the cloned zygotic embryos?
Our comment / answer
The reviewer's observation is correct.
In preliminary experiments (data not presented), different combinations of plant growth regulators (PGR) were added to the culture medium to stimulate the in vitro development of the zygotic and apomictic embryos present in the immature ovules. Interestingly, we observed that in some combinations PGR stimulated callus formation and the production of adventitious embryos from the different tissues of immature ovules. On the contrary, the PGR-free medium allowed the regular development of the zygotic and apomictic embryos already present in the immature ovules without the production of callus and adventitious embryos. Therefore, the immature ovules used in the present research were incubated on PGR-free medium to facilitate the recovery of zygotic embryos.
We have added this information in the 'Discussion' section.
Q11: Are there any known hypotheses regarding the elimination of zygotic embryos during seed/fruit maturation? Maybe the authors have their own?
Our comment / answer
We thank the reviewer for the interesting question.
It is known that in Opuntia spp., apomixis can occur mainly through sporophytic agamospermy [21-23], where adventitious embryos develop from nucellar tissue. However, development of embryos from unfertilised ovules (parthenogenesis) has also been observed in the genus Opuntia [21]. However, the exact reason for the abortion of zygotic embryos in vivo during the seed maturation processes is not known and we are focusing on this aspect in our current research. Our hypothesis is that the numerous embryos of apomictic origin which are contained in the ovule (often there are more than ten embryos per ovule) compete with the zygotic embryo by using the resources necessary for its development, causing its abortion. The increase in the proportion and number of zygotic embryos when the immature ovules are incubated on a culture medium providing sufficient nutrients, enhancing the chances of survival of zygotic embryos, substantiates this hypothesis.
We have added this information in the 'Discussion' section.
